# Comparative evaluation of TNM staging systems (eighth vs. ninth edition) for the non-surgical treatment of localized and locally advanced anal squamous cell carcinoma: Prognostic significance of T classification and lymph node status

Aihong Zheng[1☉], Hong'en Xu[2,3☉], Yiming Tao[4], Bingchen Chen[5], Jieni Ding[2], Tao Song[2]*, Yanwei Lu[2]*

1 Department of Medical Oncology, Cancer Center, Zhejiang Provincial People's Hospital, Affiliated People's Hospital, Hangzhou Medical College, Hangzhou, Zhejiang, People's Republic of China, 2 Department of Radiation Oncology, Cancer Center, Zhejiang Provincial People's Hospital, Affiliated People's Hospital, Hangzhou Medical College, Hangzhou, Zhejiang, People's Republic of China, 3 Department of Oncology, Zhejiang Provincial People's Hospital BiJie Hospital, BiJie First People's Hospital, Bijie, Guizhou, People's Republic of China, 4 Department of Interventional Medicine, Tongxiang First People's Hospital, Jiaxing, Zhejiang, People's Republic of China, 5 Division of Colorectal Surgery, Department of General Surgery, Cancer Center, Zhejiang Provincial People's Hospital, Affiliated People's Hospital, Hangzhou Medical College, Hangzhou, Zhejiang, People's Republic of China

☉ These authors contributed equally to this work.
* songtao@hmc.edu.cn (TS); 309960266@qq.com (YL)

**Data Availability Statement:** Dataset required to reproduce our study's findings are available from

## Abstract

This study aims to compare the survival discrimination of the Tumor-Node-Metastasis (TNM) eighth and ninth editions for patients with localized and locally advanced (LLA) anal squamous cell carcinoma (ASCC) treated non-surgically and to evaluate the prognostic impact of T classification and lymph node (LN) status with data from the Surveillance, Epidemiology, and End Results database. We retrospectively included 6,876 patients in the comparison. We observed the inversion of survival outcomes for stages IIB and IIIA diseases in the TNM eighth edition [median overall survival (OS): 112 months for stage IIB vs. not reached for stage IIIA]. By contrast, it demonstrated improvement in the TNM ninth edition (median OS: not reached for IIB disease vs. 120 months for IIIA disease, P<0.001). In the correlation analysis, we observed an increased correlation between T classification and TNM staging systems (r value increased from 0.78 to 0.93) and a decreased correlation for the LN status (r value decreased from 0.83 to 0.59). For OS, variable importance analysis demonstrated more weight of importance for the T classification than the LN status (0.0871 vs. 0.0048). Additionally, decision curve analysis and time-dependent receiver operating characteristic analysis confirmed the prognostic accuracy of T classification rather than the LN status. In conclusion, TNM ninth edition is a better prognostic indicator than the eighth edition for patients with LLA ASCC treated

the figshare database: Song, Tao (2024). SEER
LLA ASCC data 2004-2020. figshare. Dataset.
https://doi.org/10.6084/m9.figshare.28082384.v1.

**Funding:** This work was funded by a Grant from
the Medical Science and Technology Project of
Zhejiang Province, China (No. 2023KY050).

**Competing interests:** The authors have declared
that no competing interests exist.

non-surgically. T classification plays a more important prognostic role than the LN status
and warrants further validation.

## Introduction

Anal cancer (AC), a malignant neoplasm, arises within the anatomical structures surrounding
the anus [1, 2]. AC is a relatively rare cancer type, accounting for approximately 2.5% of all gastrointestinal malignancies [3]. Of all histological subtypes, anal squamous cell carcinoma
(ASCC) is the predominant AC type [4]. Oncologists recommend concomitant chemoradiotherapy (CRT) as the standard non-surgical treatment model for localized and locally
advanced (LLA) ASCC [5]. The seminal Radiation Therapy Oncology Group (RTOG) 0529
trial demonstrated the efficacy of CRT in patients with non-metastatic ASCC. The findings
highlighted a long-term overall survival (OS) rate of 76% at the 5-year follow-up [6]. Nevertheless, various factors, particularly the tumor stage, may affect the efficacy of CRT in the treatment of patients with LLA ASCC.

Over the past two decades, oncologists have updated the cancer staging system for ASCC
from the American Joint Committee on Cancer (AJCC) Tumor-Node-Metastasis (TNM) sixth
edition to the latest ninth edition [7–9]. Upon comparison between the AJCC TNM sixth and
seventh editions and the eighth and ninth editions, minimal changes are observed in the definition of T classification (primary tumor assessed by tumor dimension). However, significant
revisions are noted in the definition of regional lymph node (LN) metastasis in the updated
eighth and ninth editions. Specifically, the external iliac (N1b) LNs are now recognized as
regional disease sites in the latter editions. Utilizing data from the National Cancer Database
(NCDB) spanning 2014 to 2018, it has been observed that patients diagnosed with stage IIIA
(T1/T2N1M0) under the eighth edition exhibited a more favorable prognosis than those with
stage IIB (T3N0M0) [9]. This finding challenges the traditional notion that higher staging
implies a worse prognosis. Thereafter, the TNM ninth edition has introduced several adjustments, which primarily involve redefining stage IIB as T1/T2N1M0 diseases and stage IIIA as
T3N0/N1M0 diseases (Fig 1). Nevertheless, these updates warrant further validation across
datasets beyond the NCDB, ensuring a comprehensive assessment of their clinical
implications.

Building upon the aforementioned context and the findings from our previous study [10],
we aimed to conduct a retrospective study with a larger sample size, using data extracted from
the Surveillance, Epidemiology, and End Results (SEER) registry. The primary objective was to
compare the clinical utility of the eighth and ninth editions of the AJCC TNM staging systems
in patients with LLA ASCC who primarily underwent non-surgical management. Additionally, to gain a deeper understanding of the prognostic implications of T classification and LN
status in LLA ASCC, we sought to further investigate their predictive power for survival outcomes. We also planned to review well-designed, prospective clinical studies to further assess
the prognostic value of T classification and LN status on survival outcomes in the literature.

## Patients and methods

### Ethics statement

The need for obtaining written informed consent for participation in this study was not
required, given that the data within the SEER database has been de-identified and is freely

| | TNM8th edition | | TNM9th edition |
|---|---|---|---|
| Stage I | T1N0M0 | | T1N0M0 |
| Stage IIA | T2N0M0 | | T2N0M0 |
| Stage IIB | T3N0M0 | | T1/T2N1M0 |
| Stage IIIA | T1/T2N1M0 | | T3N0/N1M0 |
| Stage IIIB | T4N0M0 | | T4N0M0 |
| Stage IIIC | T3N1M0, T4N1M0 | | T4N1M0 |

**Fig 1. A brief illustration of the revisions between AJCC TNM eighth and ninth editions.**

accessible to the public upon authorization. Consequently, ethical approval from the institutional review board of Zhejiang Provincial People's Hospital was waived for this research.

## Patient selection

Data spanning 2004 to 2020 were extracted from the SEER database comprising 18 population-based registries (SEER* Stat 8.3.6), which cover approximately half of the United States population [11]. The International Classification of Diseases for Oncology, third edition (site code: C21.0–9) was used. The inclusion criteria were as follows: (1) a confirmed histopathological diagnosis of ASCC; (2) a primary diagnosis of non-metastatic ASCC; (3) age 18 years or older; and (4) patients treated with non-surgical approaches. The major exclusion criteria were as follows: (1) patients with multiple primary cancers; (2) those receiving treatments other than radiotherapy (RT) and chemotherapy (CTx); (3) survival of less than one month; and (4) cases with unclear diagnosis regarding the T classification and LN status, which could lead to misinterpretation of the TNM staging.

## Data processing

Within the original SEER database, tumor registrations were aligned with the TNM sixth edition for patients diagnosed between 2004 and 2015, the seventh edition for those diagnosed between 2016 and 2017, and the derived extent of disease 2018 T and N (2018+) for patients diagnosed from 2018 to 2020. The staging algorithm for patients diagnosed after 2018 can be accessed at: https://staging.seer.cancer.gov/eod_public/schema/3.1/anus/. Tumors were subsequently restaged according to the TNM stage categories of the eighth and ninth edition, following the staging manuals, for the purpose of survival comparisons [8, 9]. 60 years was set as the cut-off point for age comparison (less than or equal to 60 years old at diagnosis vs. more than 60 years old at diagnosis) [12, 13]. Other variables were handled based on previously published protocols [14–16].

## Determination of study outcomes

Survival data, including the survival status (alive or dead) and survival time (in months), were extracted from the SEER database. OS was determined as the duration between the day of ASCC diagnosis and that of death caused by any reason or the last recorded follow-up. Cause-

specific survival (CSS) was defined as the duration between the day of initial ASCC diagnosis and the time of death attributable to ASCC or the date of the final follow-up, whichever came first. Both OS and CSS were determined as the primary endpoints.

## Statistical analysis

We summarized the baseline data of patients with LLA ASCC using descriptive statistics and frequency tables. Kaplan-Meier estimation was utilized to plot survival curves, which were subsequently compared using the log-rank test. A higher likelihood-ratio chi-square value indicated better homogeneity in the TNM staging discrimination. We initially conducted univariate Cox proportional hazards regression analysis on the extracted variables. Subsequently, we incorporated those variables with a P-value less than 0.05, as identified in the univariate analysis, into a multivariate Cox proportional hazard regression analysis to identify independent prognostic factors for OS and CSS, The results were expressed as hazard ratios (HR) and their respective 95% confidence intervals (CIs). Furthermore, we employed the Spearman rank correlation coefficient (r) to determine the correlation between the T classification and LN status across the eighth and ninth editions of the TNM staging systems, considering an r value exceeding 0.7 as indicative of a strong correlation. Additionally, we conducted variable importance (VIMP) analysis, an intrinsic statistic of the random survival forest (RSF) algorithm, to evaluate the prognostic impact of T classification and LN status for OS and CSS. These analysis were conducted utilizing the "*corrplot*" and "*randomForestSRC*" R packages [17]. Finally, the prognostic discrimination of the T classification and LN status on OS and CSS were plotted and compared using decision curve analysis (DCAs) and the time-dependent receiver operating characteristic (tdROC) analysis with DeLong's test [15, 18].

Statistical analysis were performed via the Statistical Package for the Social Sciences software (version 25.0; IBM Corporation, Armonk, NY, USA) and R software (version 3.6.2; https://www.r-project.org, Institute for Statistics and Mathematics, Vienna, Austria). We plotted survival curves using GraphPad Prism 8.0 (GraphPad Software, San Diego, CA, USA). P-values < 0.05 indicated statistical significance.

## Results

### Patient characteristics

The final analysis consisted of 6,876 patients with LLA ASCC (Fig 2). Table 1 depicts the baseline characteristics of all patients. The median age at diagnosis was 59 years, whereas 3,521 patients (51.2%) were diagnosed with LLA ASCC after 60 years. Caucasians accounted for 87.7% of the patients, and there were 4,865 (70.8%) women. Approximately half of the patients (47.0%) were diagnosed with T2, whereas 2,809 (40.9%) patients were positive for LNs. According to the TNM eighth edition, 874 (12.7%), 2,302 (33.5%), 581 (8.4%), 1,117 (16.2%), 310 (4.5%), and 1,692 (24.7%) patients were diagnosed with stages I (T1N0M0), IIA (T2N0M0), IIB (T3N0M0), IIIA (T1-2N1M0), IIIB (T4N0M0), and IIIC (T3-4N1M0), respectively. According to the TNM ninth edition, 1,117 (16.2%), 1,761 (25.6%), and 512 (7.5%) patients were diagnosed with stage IIB (T1-2N1M0), IIIA (T3N0-1M0), and IIIC (T4N1M0), respectively. The proportion of patients in other stages remained unchanged between the eighth and ninth editions.

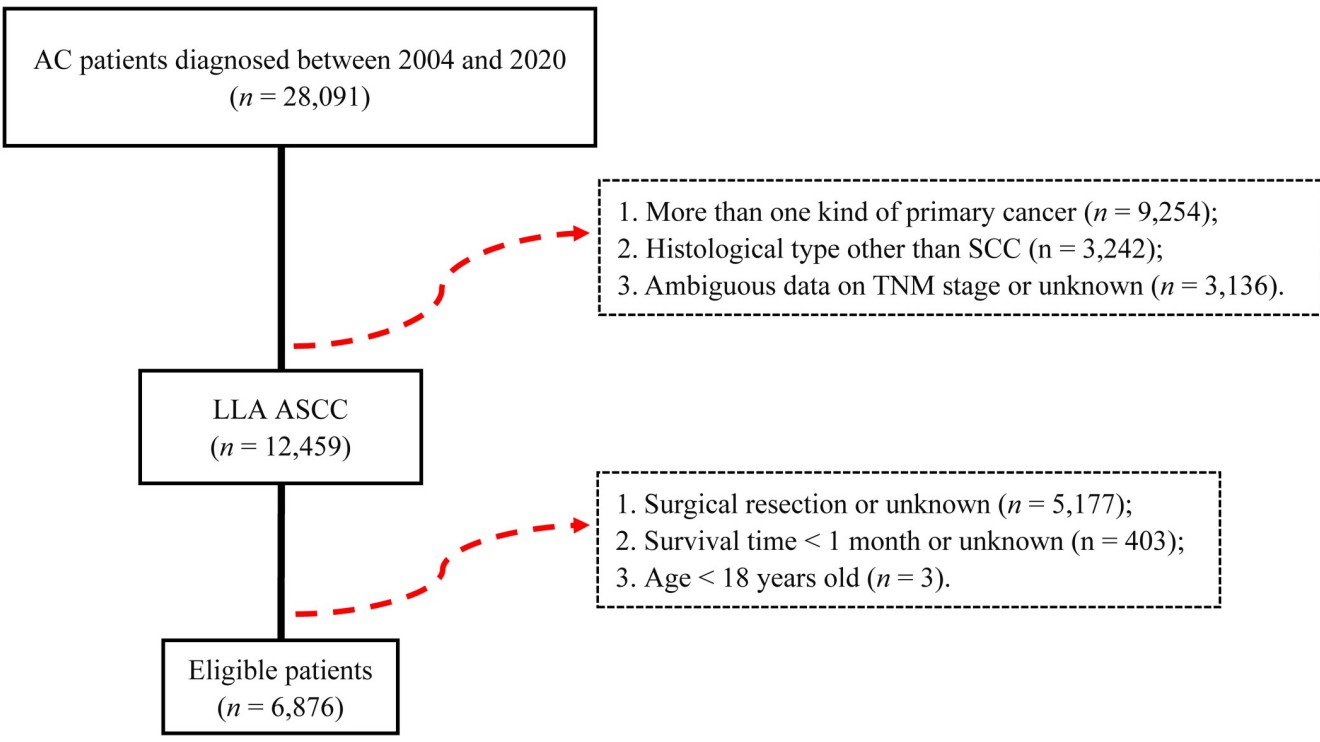

**Fig 2. Patient selection flowchart.**

### Survival discrimination and multivariate analysis

Fig 3 illustrates the OS and CSS curves based on the TNM eighth and ninth editions. Overall, the 5- and 10-year OS rates were 68.3% (95% CI, 0.671–0.695) and 57.1% (95% CI, 0.555–0.587), respectively. The 5- and 10-year CSS rates were 76.6% (95% CI, 0.754–0.778) and 72.5% (95% CI, 0.711–0.739), respectively. The median OS time was 168 months, and the median CSS time was not reached at the last follow-up.

In the TNM eighth edition, we observed a paradoxical relationship between stage IIB and IIIA diseases, with a median OS of 112 months for stage IIB compared to not reached for stage IIIA ($\chi^2$ = 27.5, P < 0.001; Fig 3A). Additionally, an inversion was noted between stage IIIB and IIIC diseases, with median OS of 77 months and 100 months, respectively ($\chi^2$ = 3.314, P = 0.069). For CSS, a similar trend was observed between stage IIB and IIIA diseases ($\chi^2$ = 16.813, P < 0.001; Fig 3B).

In the updated TNM ninth edition for OS (Fig 3C), we found improved survival discrimination between stage IIB and IIIA diseases, with a median OS of not reached for stage IIB and 120 months for stage IIIA ($\chi^2$ = 29.758, P < 0.001). However, no significant difference was identified between stage IIIB and IIIC diseases ($\chi^2$ = 1.541, P = 0.214; Fig 3C). Regarding CSS, the survival differences were highly significant among all subgroups (P < 0.001), except for the difference between stage IIIB and IIIC diseases, which demonstrated a marginally less pronounced distinction (P = 0.021). S1 Table summarizes the results of univariate and multivariate Cox regression analysis. Both TNM eighth and ninth editions were independent prognostic factors correlated with OS and CSS (all P < 0.001).

**Table 1. Baseline clinical and treatment features of patients with LLA ASCC.**

| Characteristic | Frequency (%) |
|---|---|
| *Age at diagnosis (years)* | |
| $\leq 60$ | 3355 (48.8) |
| $> 60$ | 3521 (51.2) |
| *Sex* | |
| Female | 4865 (70.8) |
| Male | 2011 (29.2) |
| *Race* | |
| White | 6032 (87.7) |
| Non-white | 844 (12.3) |
| *T stage* | |
| $T_{0-1}$ | 1059 (15.4) |
| $T_2$ | 3234 (47.0) |
| $T_3$ | 1761 (25.6) |
| $T_4$ | 822 (12.0) |
| *N stage* | |
| Negative | 4067 (59.1) |
| Positive | 2809 (40.9) |
| *AJCC TNM8th edition* | |
| I | 874 (12.7) |
| IIA | 2302 (33.5) |
| IIB | 581 (8.4) |
| IIIA | 1117 (16.2) |
| IIIB | 310 (4.5) |
| IIIC | 1692 (24.7) |
| *AJCC TNM9th edition* | |
| I | 874 (12.7) |
| IIA | 2302 (33.5) |
| IIB | 1117 (16.2) |
| IIIA | 1761 (25.6) |
| IIIB | 310 (4.5) |
| IIIC | 512 (7.5) |
| *RT* | |
| No/Unknown | 376 (5.5) |
| Yes | 6500 (94.5) |
| *CTx* | |
| No/Unknown | 621 (9.0) |
| Yes | 6255 (91.0) |

## Analysis of the T classification and LN status

The correlation analysis, as illustrated in Fig 4A, revealed an enhanced correlation of the T classification from the eighth to the ninth edition of the TNM staging system (r value increased from 0.78 to 0.93). In contrast, the LN status showed a decreased correlation between the eighth and ninth editions (r value decreased from 0.83 to 0.59).

Next, we employed VIMP analysis to evaluate the prognostic impact of the T classification and LN status on OS and CSS. The T classification demonstrated a more substantial prognostic influence compared to the LN status, with importance rates (IR) of 8.71% for T classification

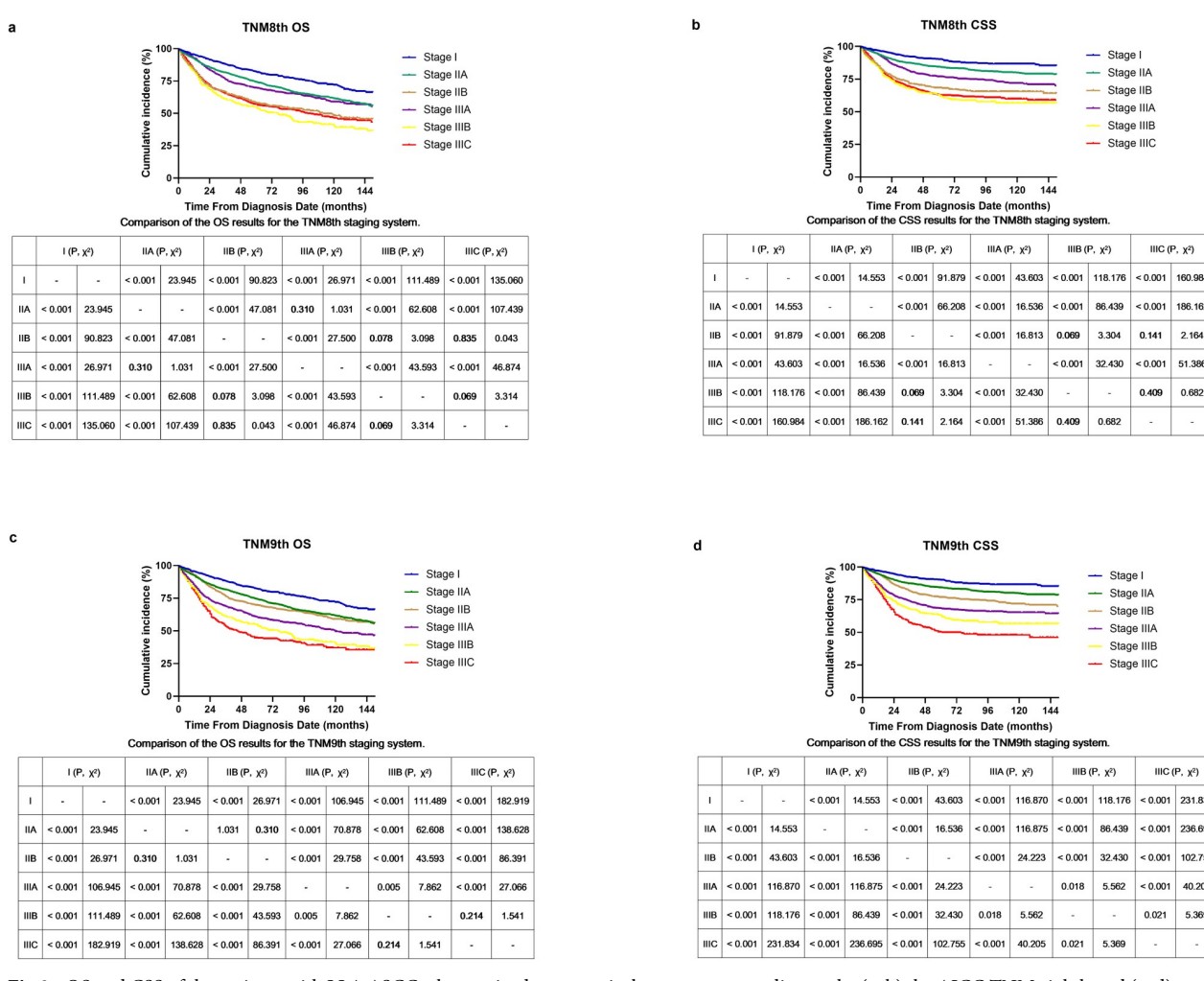

**Fig 3.** OS and CSS of the patients with LLA ASCC who received non-surgical treatment according to the (a, b) the AJCC TNM eighth and (c, d) ninth editions.

versus 0.48% for LN status in OS, and 9.20% for T classification versus 1.43% for LN status in CSS (Fig 4B).

We further conducted Kaplan–Meier survival curves and multivariate Cox proportional hazards regression analysis to assess the impact of T classification and LN status on OS and CSS. Both T classification and LN status exhibited strong discriminatory power, as shown by the Kaplan–Meier survival curves (S1 Fig). Additionally, both variables were identified as independent prognostic indicators with significant associations in the multivariate analysis (all P < 0.001; S2 Table).

Finally, DCAs indicated that the T classification provided a greater net clinical benefit for predicting OS (Fig 4C) and CSS (Fig 4D) across all threshold probabilities compared to the LN status. We also compared the accuracy of T classification and LN status in predicting 5- and 10-year OS and CSS using tdROC analysis. The T classification consistently demonstrated superior predictive power compared to the LN status, with the semi-parametric DeLong's test confirming these differences (all P < 0.001) (S2 Fig).

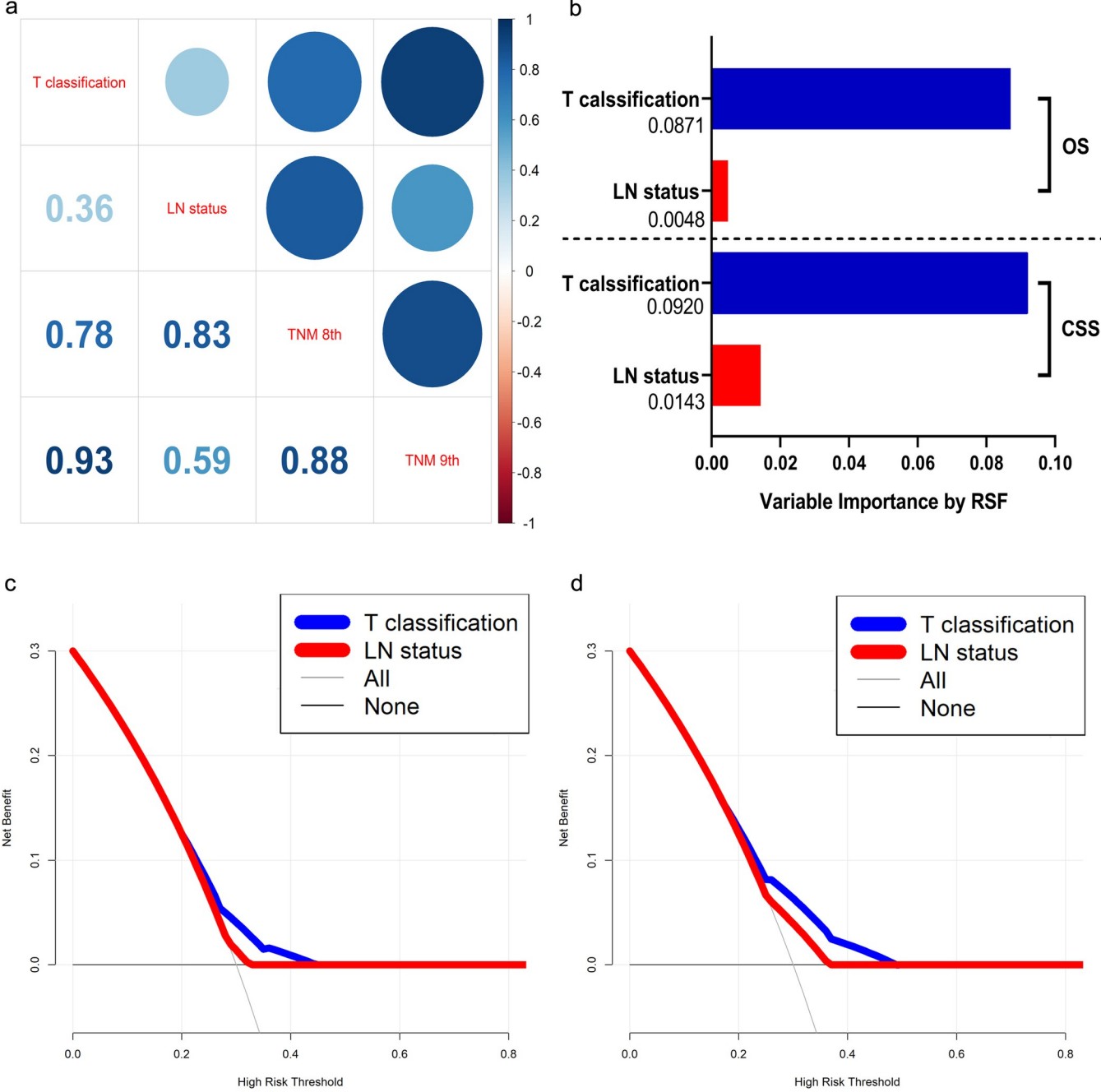

**Fig 4.** a: Correlation coefficients between the T classification, LN status, and TNM eighth and ninth editions; b: Variable importance analysis by random survival forest (RSF) algorithm among the T classification, LN status, OS, and CSS; c: Decision curve analysis of OS between T classification and LN status; d: Decision curve analysis of CSS between T classification and LN status.

## Discussion

Previous studies have demonstrated favorable survival outcomes in patients with LLA ASCC who underwent combined RT and CTx, with acceptable treatment-related sequelea [19]. This has led to heightened expectations for the application of the widely accepted TNM staging system to facilitate more precise tumor stratification [10]. To address this research gap, we re-

engaged the SEER database with updated data to better meet the needs of clinical practice. Our results indicate that the ninth edition of the TNM staging system offers greater precision for patients with LLA ASCC primarily managed non-surgically, complementing the findings from the NCDB. Furthermore, we explored the predictive power of two key components: T classification and LN status. Through various statistical methods, our findings suggest that T classification is a more significant predictor of survival outcomes than LN status, which contrasts with previous studies. A comprehensive summary and discussion of the prognostic impact of T classification and LN status will presented subsequently (Table 2).

LN categories in the previous TNM sixth and seventh editions have been modified in the subsequent editions [7, 8]. The newer editions have introduced definitions for N1a, N1b, and N1c, where the involvement of N1b LNs indicates a regional disease [9]. Compared with the TNM eighth edition (Fig 1), the implication of T classification was enhanced by upstaging T1/T2 diseases with positive LNs to stage IIB in the ninth edition. Additionally, T3 disease was consolidated uniformly into stage IIIA regardless of their LN status [20]. The Kaplan–Meier survival curves for OS and CSS in Fig 3. clearly demonstrate that the TNM ninth edition provides improved survival discrimination for LLA ASCC compared to the eighth edition. To further investigate the impact of staging changes on survival outcomes, we initially conducted correlation analysis, which showed that the correlation between T classification and tumor staging systems is gradually strengthening (correlation coefficient increased from 0.78 to 0.93), while the correlation of LN status is diminishing (from 0.83 to 0.59). Subsequently, using VIMP, we determined that the IR of T classification is greatly higher than that of LN status for both OS (8.71% versus 0.48%) and CSS (9.20% versus 1.43%). Ultimately, DCAs and tdROC

**Table 2. Landmark prospective studies on the impact of T classification and LN involvement on the survival of AC.**

| Trial ID (reference) | Year | No. of patients / Study arm | Tumor stage | Main results on the impact of tumor classification and LN involvement. |
|---|---|---|---|---|
| RTOG 87-04/ ECOG 1289 [21] | 1996 | 291 (RT+5-Fu vs. RT+5-Fu/MMC) | LLA AC | The addition of MMC significantly benefited CFR in RTOG T3-4 diseases, but not in RTOG T1-2 diseases; The addition of MMC benefited CFR in LN-positive patients without significant difference. |
| UKCCCR [22] | 1996 | 577 (RT alone vs. RT+5-Fu/MMC) | > 95% LLA AC | T4 and LN involvement decreased the CSS. |
| EORTC 22861 [23] | 1997 | 103 (RT alone vs. RT+5-Fu/MMC) | LA AC | Tumor classification had no prognostic value on survival outcomes; Positive LN status significantly impaired both local control and OS. |
| | | RT boost might consider after 6 weeks | | |
| KANAL 2 ACCORD 03 [24] | 2001 2012 | 80 (ICT followed by HD RT +5-Fu/CDDP) 307 (± ICT followed by CRT: HD vs. SD) | LASCC | Beyond RTOG T1 diseases and involved inguinal LNs significantly associated with a decreased CFS. |
| RTOG 92-08/ RTOG 87–04 (MMC) [21, 25] | 2008 | 20 (RT+5-Fu/MMC) 148 (RT+5-Fu/MMC) | LLA AC | Large tumor diameter (≥ 5cm) significantly decreased LRF and OS. |
| RTOG 98–11 [26, 27] | 2010 2012 | 644 (RT+5-Fu/MMC vs. RT +5-Fu/CDDP) | LLA AC | Large tumor diameter (> 5cm) and positive LN status significantly decreased DFS and OS. |
| ACI 1 [28] | 2013 | 283 (RT+5-Fu/MMC) | LLA AC | Positive LN status significantly impaired LRF, CSS and OS. |
| ACT II [29] | 2013 | 940 (CRT: CDDP vs. MMC) ± CTx maintenance | LLA SCC | PFS: 80–84% for T1-2 diseases and 62–67% for T3-4 diseases across different arms (all P > 0.05); PFS: 68% for LN-positive patients and 76% for LN-negative patients, with no difference. |

5-FU: 5- fluorouracil; MMC: mitomycin C; CDDP: cisplatin; ICT: induction chemotherapy; HD: high-dose; SD: standard-dose; RT: radiotherapy; RCT: radiochemotherapy; LLA: localized and locally advanced; SCC: squamous cell carcinoma; AC: anal cancer; LRF: localregional failure; DFS: disease-free survival.

confirmed the superior prognostic accuracy of T classification over LN status (all P < 0.001). These results collectively demonstrate that the ninth edition of the TNM staging system exhibits greater prognostic accuracy compared to its eighth edition.

We then summarized a series of seminal, high-level evidence of clinical trials focusing on the prognostic influence of T classification and LN status chronologically (Table 2) [21–29]. In the European Organization for Research and Treatment of Cancer 22861 trial [23], patients with clinically positive LNs predicted not only poor OS (P = 0.045) but also decreased loco-regional control (P = 0.0017). By contrast, T classification exerted no prognostic effect on the survival outcomes. Similarly, post-hoc analysis from the first UKCCCR Trial indicated that positive LNs were a stronger prognostic indicator for higher locoregional failure (HR = 1.87, P = 0.012), lower CSS (HR = 1.83, P = 0.031), and OS (HR = 1.74, P = 0.006) respectively [28]. These phenomena warrant investigating the generalizability of the findings. In addition to the negative influence of LNs on prognosis (ACT-II trial) [29], the RTOG 92–08 and 98–11 trials suggest that larger tumor size is an independent risk factor significantly affecting patient survival [25–27]. Another large retrospective study from NCDB extracted data from 19,199 patients with ASCC [30]. A multivariable analysis using the Cox regression model suggested that T classification exerted an independent prognostic impact on the OS (all P < 0.001). A 2021 SEER study reported that tumor size exerted a significant prognostic value on the OS in 2,458 patients with stages I to IV AC. However, the cut-off value of the tumor size did not adhere to the T-stage definition in the TNM staging manual [31].

We focused exclusively on patients with LLA ASCC who underwent non-surgical treatment. Consequently, all patients were staged clinically rather than pathologically [32]. This aspect highlights the growing implication for radiation oncologists to employ radiological examinations to attain more precise tumor staging. Among various imaging modalities, magnetic resonance imaging (MRI) is advantageous due to its high tissue resolution and is recommended in major guidelines [33–35]. A survey of expert opinions in the USA supported the use of MRI, with over 90% of respondents advocating for its application in primary T classification. However, the detection of metastatic regional LNs remains controversial [36]. Relying solely on size criteria for detecting metastatic LNs based on the short-axis dimension can be error-prone [37], potential causes of these errors include partial visualization of inguinal LNs and missed detection of metastatic LNs due to improper placement of the saturation band applied in abdominal MRI [38]. In this context, positron emission tomography/computed tomography (PET/CT) has demonstrated superior diagnostic ability for detecting metastatic LNs compared to CT and MRI [39]. However, the prognostic influence of metastatic LNs identified by radiological findings deserves in-depth analysis. In this aspect, given the partial similarity in lymphatic drainage, a randomized controlled trial (Uterus-11) on cervical cancer showed that laparoscopic staging led to upstaging in 39 of 120 (33%) patients. Nonetheless, no significant difference in disease-free survival was observed between surgical and clinical staging methods for patients with stage IIB–IVA cervical cancer who underwent CRT [40].

This study has some limitations. First, we performed a retrospective analysis with data extracted from the SEER database, despite an increased sample size over our previous study [10], further validation in required. Secondly, the AJCC TNM manual defined metastasis to the external iliac LNs as distant metastasis in patients diagnosed with the sixth and seventh editions of TNM, which could have introduced bias in the final interpretation. Fortunately, the RTOG has previously recommended the use of a contouring atlas for AC and suggested delineating the drainage area of regional LNs, including the external iliac LNs, for irradiation, which could moderately reduce the bias presented in this study [41]. Finally, crucial parameters, such as the human papillomavirus (HPV) infection status, general performance status, and tumor marker levels, were unavailable. Furthermore, treatment-related data from the

SEER database were inaccessible. A de-escalation treatment strategy has been employed successfully for patients with head and neck squamous cell carcinoma who tested positive for HPV. Therefore, researchers should investigate the potential application of this approach for ASCC treatment in the future.

## Conclusions

In this study, we compared the clinical usefulness of the TNM eighth and ninth editions in patients with LLA ASCC who primarily underwent non-surgical treatment. The TNM ninth edition resulted in superior OS and CSS discrimination than the eighth edition. Furthermore, the T classification demonstrated a more prognostic importance weight than the LN status, which can provide a direction for improvements in AC staging in the future.

## Supporting information

**S1 Fig.** OS and CSS of patients with LLA ASCC who received non-surgical treatment according to the (a, b) T classification and (c, d) LN status.
(TIF)

**S2 Fig.** tdROC curves of the T classification and LN status at (a, b) 5-year OS and CSS, and (c, d) 10-year OS and CSS.
(TIF)

**S1 Table. Univariate and multivariate Cox regression analysis of OS and CSS with TNM eighth and ninth editions.**
(DOCX)

**S2 Table. Univariate and multivariate Cox regression analysis of OS and CSS with the T classification and LN status.**
(DOCX)

## Author Contributions

**Conceptualization:** Aihong Zheng, Hong'en Xu, Tao Song, Yanwei Lu.

**Data curation:** Aihong Zheng, Tao Song, Yanwei Lu.

**Formal analysis:** Aihong Zheng, Tao Song, Yanwei Lu.

**Funding acquisition:** Tao Song.

**Investigation:** Hong'en Xu, Yiming Tao.

**Methodology:** Hong'en Xu, Yiming Tao, Bingchen Chen, Jieni Ding.

**Project administration:** Yiming Tao, Bingchen Chen.

**Resources:** Yiming Tao, Bingchen Chen, Jieni Ding, Tao Song.

**Software:** Tao Song.

**Supervision:** Aihong Zheng, Bingchen Chen, Tao Song, Yanwei Lu.

**Validation:** Hong'en Xu, Bingchen Chen, Tao Song, Yanwei Lu.

**Visualization:** Hong'en Xu, Jieni Ding, Tao Song, Yanwei Lu.

**Writing – original draft:** Aihong Zheng, Hong'en Xu, Yiming Tao, Bingchen Chen, Jieni Ding, Tao Song, Yanwei Lu.

**Writing – review & editing:** Aihong Zheng, Hong'en Xu, Yiming Tao, Bingchen Chen, Jieni Ding, Tao Song, Yanwei Lu.

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
