## [Decision Letter · Decision Letter 0]

8 Nov 2024

PONE-D-24-28181Comparative Evaluation of TNM Staging Systems (eighth vs. ninth version) for the Non-Surgical Treatment of Localized and Locally Advanced Anal Squamous Cell Carcinoma: Prognostic Significance of T Classification and Lymph Node Status.PLOS ONE

Dear Dr. Song,

Thank you for submitting your manuscript to PLOS ONE. After careful consideration, we feel that it has merit but does not fully meet PLOS ONE’s publication criteria as it currently stands. Therefore, we invite you to submit a revised version of the manuscript that addresses the points raised during the review process.

We look forward to receiving your revised manuscript.

Kind regards,

Tsutomu Kumamoto

Academic Editor

PLOS ONE

**Journal Requirements:**

This work was funded by a Grant from the Medical Science and Technology Project of Zhejiang Province, China (No. 2023KY050).

3. In the online submission form, you indicated that The datasets used and analyzed are included in the article. Further inquiries can be directed to the corresponding author on reasonable request.

**Additional Editor Comments:**

The authors have conducted an excellent study evaluating the validity of the TNM classification for AACS. However, there are several improvements that can be made.

1. Abbreviations such as VIA, DCAs, and tdROCs are used in the abstract.

2. It is necessary to describe how the cT and cN stages were determined in the methods section.

3. The subtitle for Figure 2 with n=15,298 should be included. Since exclusion criteria are listed in two stages, there must be a corresponding reason.

4. As mentioned in the first paragraph of the discussion, the strengths of this study are highlighted through various analyses. It is important to adequately describe what these analyses mean for the readers.

5. The second paragraph of the discussion mentions the validity of the change from the 8th to the 9th edition. What kind of research background led to this revision? It would be good to include references as well."

Reviewers' comments:

Reviewer's Responses to Questions

**Comments to the Author**

1. Is the manuscript technically sound, and do the data support the conclusions?

Reviewer #1: Yes

Reviewer #2: Yes

2. Has the statistical analysis been performed appropriately and rigorously? 

Reviewer #1: Yes

Reviewer #2: Yes

3. Have the authors made all data underlying the findings in their manuscript fully available?

Reviewer #1: Yes

Reviewer #2: Yes

4. Is the manuscript presented in an intelligible fashion and written in standard English?

Reviewer #1: Yes

Reviewer #2: Yes

5. Review Comments to the Author

**Reviewer #1:** I want to congratulate you on writing this manuscript. It is very well structured and contributes to the field.

I have few minor suggestions.

1. In the abstract what does VIA stand for?

2. In the abstract what do the the abbreviations “DCA” and “tdROC” stand for? It's written in the text but should be written in the abstract too.

3. In Table 2 “RCT: radiotherapy” should be “RCT: radiochemotherapy”; “LRF:localregional failure”should be “LRF:locoregional failure”.

4. The 1st paragraph of the discussion “treatment-related sequels” should be “treatment-related sequelea”.

5. You clearly defined your study's limitations, but you should also define its strengths. In my opinion, the strength lies in the absence of a direct comparison between the TNM 8th and 9th versions for anal cancer in the literature, making this study valuable. It proved the clinical usefulness of the TNM 9th version in a SEER database besides the NCDB. It also highlighted that T status is more important than N status for survival, in contrast to other studies.

**Reviewer #2: **Introduction:

It's better to briefly summarize the principal differences between the eighth and ninth versions of TNM. Identify limitations of earlier studies to point out the novelty of your study.

Patients and methods:

It will be beneficial to Provide more details of inclusion/exclusion criteria for patients and re-organization of data according to TNM version. Justify the choice of statistical methods used (the Cox proportional hazard regression model)

Results:

Can you Simplify and present most complex statistical findings in a manner that is understandable for reader (Include more interpretation of important metrics, such as correlation coefficients and survival curves).

Discussion:

Explain more why the ninth edition of TNM is more prognostically accurate. More literature should be integrated to compare the findings of the study against existing research. Provide some possible explanations for the overall lower predictive value of lymph node status.

6. PLOS authors have the option to publish the peer review history of their article (what does this mean?). If published, this will include your full peer review and any attached files.

Reviewer #1: No

Reviewer #2: No

---

## [Author Response · Author response to Decision Letter 0]

23 Dec 2024

Point-by-point responses to the reviewers and editor

First, we would like to thank the reviewers and editor for their constructive and positive comments on our manuscript.

Replies to Reviewer 1

I want to congratulate you on writing this manuscript. It is very well structured and contributes to the field.

I have few minor suggestions.

1. In the abstract what does VIA stand for?

2. In the abstract what do the the abbreviations “DCA” and “tdROC” stand for? It's written in the text but should be written in the abstract too.

3. In Table 2 “RCT: radiotherapy” should be “RCT: radiochemotherapy”; “LRF:localregional failure”should be “LRF:locoregional failure”.

4. The 1st paragraph of the discussion “treatment-related sequels” should be “treatment-related sequelea”.

Answer: Thank you for your comment. We have revised the issues mentioned above in the manuscript, and once again, we are grateful for your comments.

5. You clearly defined your study's limitations, but you should also define its strengths. In my opinion, the strength lies in the absence of a direct comparison between the TNM 8th and 9th versions for anal cancer in the literature, making this study valuable. It proved the clinical usefulness of the TNM 9th version in a SEER database besides the NCDB. It also highlighted that T status is more important than N status for survival, in contrast to other studies.

Answer: We highlighted the clinical significance of the present study in the first paragraph of the “Discussion” section based on your suggestion. 

Thank you again for your hard work in reviewing our manuscript.

Replies to Reviewer 2

Introduction:

It's better to briefly summarize the principal differences between the eighth and ninth versions of TNM. Identify limitations of earlier studies to point out the novelty of your study.

Answer: Thank you for your comment. We have updated our description on the differences between the TNM eighth and ninth editions for anal carcinoma and highlighted the clinical value of the present study in paragraphs 2 and 3 of the “Introduction” section based on your suggestion.

Patients and methods:

It will be beneficial to Provide more details of inclusion/exclusion criteria for patients and re-organization of data according to TNM version. Justify the choice of statistical methods used (the Cox proportional hazard regression model)

Answer: We have thoroughly revised the methodology section, with particular emphasis on the statistical methods in response to your concerns. In the revised manuscript, we have detailed the methods employed in both univariate and multivariate Cox proportional hazards regression analysis, as well as the re-organization of data according to different TNM editions. Additionally, we have refined the description of the inclusion and exclusion criteria based on your comment. To address your follow-up inquiry regarding the interpretation of the statistical results in the results section, we have included a reference that employs similar methodologies (JAMA Netw Open. 2023 May 1;6(5):e2312022). This addition is intended to enhance the readers' understanding and interpretation of the presented results. We quite appreciate for your kindly comments.

Results:

Can you Simplify and present most complex statistical findings in a manner that is understandable for reader (Include more interpretation of important metrics, such as correlation coefficients and survival curves).

Answer: Thank you for your comment. We have thoroughly revised the interpretation of the “Analysis of the T classification and LN status” and the interpretation of survival curves within the results section to improve manuscript clarity.

Discussion:

Explain more why the ninth edition of TNM is more prognostically accurate. More literature should be integrated to compare the findings of the study against existing research. Provide some possible explanations for the overall lower predictive value of lymph node status.

Answer: Thank you for your comment. The ninth edition of the TNM staging system for anal carcinoma, based on 2014-2018 data from the National Cancer Database (NCDB), has revealed that patients classified as stage IIIA (T1/T2N1M0) under the eighth edition have a more favorable prognosis than those at stage IIB (T3N0M0), which challenges conventional clinical expectations. This new staging system has been robustly validated using NCDB data, as detailed in the literature (CA Cancer J Clin. 2023;73(5):516-23). To further substantiate these findings, we also utilized the SEER database, focusing on localized and locally advanced (LLA) anal squamous cell carcinoma (ASCC) patients who received concurrent chemoradiotherapy (CRT), and the results corroborate the NCDB outcomes.

Furthermore, our analysis of the predictive performance of T classification and lymph node (LN) status in LLA ASCC patients involved several statistical methodologies. Initially, we conducted correlation analysis, which showed that the correlation between T classification and tumor staging systems is gradually strengthening, while the correlation of LN status is diminishing. Subsequently, using variable importance analysis, an intrinsic statistic of the random survival forest algorithm, we determined that the importance of T classification is greatly higher than that of LN status for both OS and CSS. Additionally, Kaplan-Meier survival analysis and univariate and multivariate Cox regression analysis identified both T classification and LN status as independent prognostic factors. Finally, decision curve analysis and time-dependent ROC analysis confirmed the superior prognostic accuracy of T classification over LN status. These statistical methodologies provide strong evidence that the ninth edition of the TNM staging system is more prognostically accurate than the eighth edition.

Given that anal carcinoma is a relatively rare cancer type, studies with small sample sizes have limited value for tumor stage comparisons. Therefore, we summarized landmark prospective studies in the discussion section, focusing on the impact of T classification and LN status on the survival of anal carcinoma. Contemporary studies, such as the RTOG series (RTOG 92-08 and RTOG 98-11), support advanced T classification as a clinically significant factor affecting patient prognosis. Currently, CRT is the standard treatment for LLA ASCC, making the use of medical imaging to determine T classification and LN status crucial. Among these, MRI is advantageous due to its high tissue resolution and is recommended in major guidelines. However, the detection of metastatic regional LNs remains controversial. Relying solely on size criteria for detecting metastatic LNs on the short axis can be error-prone (Eur Radiol. 2011 Apr;21(4):776-85, and Korean J Radiol. 2017 Nov-Dec;18(6):946-956), with potential partial visualization of inguinal LNs and missed detection of metastatic LNs due to improper placement of the saturation band used in abdominal MRI (Abdom Radiol (NY). 2019;44:3726-3739). In this context, positron emission tomography (PET/CT) has demonstrated superior diagnostic ability for detecting metastatic LNs compared to CT and MRI (Abdom Radiol (NY). 2024 May;49(5):1351-1362). However, the prognostic influence of metastatic LNs identified by radiological findings deserves in-depth analysis. In this aspect, given the partial similarity in lymphatic drainage, a randomized controlled trial (Uterus-11) on cervical cancer showed that laparoscopic staging led to upstaging in 39 of 120 (33%) patients. Nonetheless, no significant difference in disease-free survival was observed between surgical and clinical staging methods for patients with stage IIB–IVA cervical cancer who underwent CRT (Int J Gynecol Cancer. 2020;30:1855-1861).

We have revised our discussion section to incorporate the aforementioned references, thereby enhancing our exploration of the relatively lower predictive value of LN status. Further validation through large-scale, prospective clinical studies is necessary to confirm these findings.

Thank you again for your hard work in reviewing our manuscript.

Journal Requirements:

Please ensure that your manuscript meets PLOS ONE's style requirements, including those for file naming. The PLOS ONE style templates can be found at.

Answer: Thank you for your comment. We have revised our manuscript following the PLOS ONE’s style requirements. 

This work was funded by a Grant from the Medical Science and Technology Project of Zhejiang Province, China (No. 2023KY050).

Answer: We confirm that the funder mentioned in the present study had no role in study design, data collection and analysis, decision to publish, or preparation of the manuscript. We have added this statement in the cover letter. Thank you for your kindly suggestion.

3. In the online submission form, you indicated that The datasets used and analyzed are included in the article. Further inquiries can be directed to the corresponding author on reasonable request.

Answer: We have revised our statement regarding “Data Availability” in the online submission system: Publicly available datasets were analyzed in this study. These data can be found here: https://seer.cancer.gov/. The datasets analyzed are included within the manuscript and its supporting information files.

Answer: Thank you for your comment. We have revised our ethics statement based on your suggestion.

Additional Editor Comments:

The authors have conducted an excellent study evaluating the validity of the TNM classification for AACS. However, there are several improvements that can be made.

1. Abbreviations such as VIA, DCAs, and tdROCs are used in the abstract.

Answer: We have revised our abstract based on your concern.

2. It is necessary to describe how the cT and cN stages were determined in the methods section.

Answer: We appreciate your comment. Accordingly, we have updated the "Data Processing" section to explicitly outline the procedures for determining T classification and LN status, adhering to your recommendation.

3. The subtitle for Figure 2 with n=15,298 should be included. Since exclusion criteria are listed in two stages, there must be a corresponding reason.

Answer: We have revised our Figure 2 following on your kindly suggestion.

4. As mentioned in the first paragraph of the discussion, the strengths of this study are highlighted through various analyses. It is important to adequately describe what these analyses mean for the readers.

Answer: Thank you for your comment. To further investigate the impact of staging changes on OS and CSS, we initially conducted correlation analysis, which showed that the correlation between T classification and tumor staging systems is gradually strengthening, while the correlation of LN status is diminishing. Subsequently, using variable importance analysis, we determined that the importance of T classification is significantly higher than that of LN status. Ultimately, decision curve analysis and time-dependent ROC analysis confirmed the superior prognostic accuracy of T classification over LN status. These statistical methodologies provide strong evidence that the ninth edition of the TNM staging system is more prognostically accurate than the eighth edition. In response to your subsequent question and concerns raised by another reviewer, we have thoroughly revised the discussion section.

5. The second paragraph of the discussion mentions the validity of the change from the 8th to the 9th edition. What kind of research background led to this revision? It would be good to include references as well."

Answer: We appreciate your comment. In the second paragraph of the "Introduction" section, we noted that in daily clinical practice, patients diagnosed with stage IIIA (T1/T2N1M0) under the eighth edition of the TNM staging system exhibited a more favorable prognosis compared to those with stage IIB (T3N0M0). This finding challenges the traditional notion that higher staging implies a worse prognosis. In light of this and considering the need to optimize the stage III subgroups (T3 diseases), the ninth edition of the TNM staging system has made moderate improvements (Figure 1).

Thank you for your kindly suggestion. We have incorporated a new reference, number 20 (Ann Surg Oncol. 2024;31(7):4155-8), in the second paragraph of the "Discussion" section. Additionally, integrating suggestions from another reviewer, we have expanded our exploration of the relatively lower predictive value of LN status in the fourth paragraph of the discussion and included several key references. We hope that these revisions will enhance the readers' understanding and interpretation of the presented results.

Thank you again for your hard work in handling our manuscript.

---

## [Editor Report · Decision Letter 1]

2 Jan 2025

Comparative Evaluation of TNM Staging Systems (eighth vs. ninth edition) for the Non-Surgical Treatment of Localized and Locally Advanced Anal Squamous Cell Carcinoma: Prognostic Significance of T Classification and Lymph Node Status.

PONE-D-24-28181R1

Dear Dr. Aihong Zheng,

We’re pleased to inform you that your manuscript has been judged scientifically suitable for publication and will be formally accepted for publication once it meets all outstanding technical requirements.

Kind regards,

Tsutomu Kumamoto

Academic Editor

PLOS ONE

---

## [Editor Report · Acceptance letter]

7 Jan 2025

PONE-D-24-28181R1 

PLOS ONE

Dear Dr. Song, 

I'm pleased to inform you that your manuscript has been deemed suitable for publication in PLOS ONE. Congratulations! Your manuscript is now being handed over to our production team.

Kind regards, 

on behalf of

M.D., Ph.D. Tsutomu Kumamoto 

Academic Editor

PLOS ONE